# Grimace Scores: Tools to Support the Identification of Pain in Mammals Used in Research

**DOI:** 10.3390/ani10101726

**Published:** 2020-09-23

**Authors:** Shari Cohen, Thierry Beths

**Affiliations:** 1Office of Research, Ethics, and Integrity, University of Melbourne, Melbourne 3010, Australia; 2Anesthesia and Veterinary Clinical Sciences, University of Melbourne, Melbourne 3010, Australia; thierry.beths@unimelb.edu.au

**Keywords:** grimace scores, pain, laboratory animals, pain assessment

## Abstract

**Simple Summary:**

The ability to identify and assess pain is paramount in animal research to address the ‘refinement’ principle of the 3Rs (Reduction, Refinement, Replacement), satisfy public acceptability of animal use in research and address ethical and legal obligations. Many physiological, behavioural and physical pain assessments are commonly used, but all have their limitations. Grimace scales are a promising adjunctive behavioural pain assessment technique in some mammalian species used in research. This paper reviews the extant literature studying pain assessment techniques in general, and grimace scales specifically, in animal research. The results indicate that the grimace scale technique is simple and able to be used spontaneously at the ‘cage side’, is non-invasive in its application, highly repeatable, reliable between interobserver and intraobserver applications and easy to train and use. The use of grimace scales should be more frequently considered as an important parameter of interest in research and animal wellbeing. Further research into the use of grimace scales is required to develop scales for a wider range of animal species, increase applicability in studies specifically related to pain assessment and for further validation of the technique.

**Abstract:**

The 3Rs, Replacement, Reduction and Refinement, is a framework to ensure the ethical and justified use of animals in research. The implementation of refinements is required to alleviate and minimise the pain and suffering of animals in research. Public acceptability of animal use in research is contingent on satisfying ethical and legal obligations to provide pain relief along with humane endpoints. To fulfil this obligation, staff, researchers, veterinarians, and technicians must rapidly, accurately, efficiently and consistently identify, assess and act on signs of pain. This ability is paramount to uphold animal welfare, prevent undue suffering and mitigate possible negative impacts on research. Identification of pain may be based on indicators such as physiological, behavioural, or physical ones. Each has been used to develop different pain scoring systems with potential benefits and limitations in identifying and assessing pain. Grimace scores are a promising adjunctive behavioural technique in some mammalian species to identify and assess pain in research animals. The use of this method can be beneficial to animal welfare and research outcomes by identifying animals that may require alleviation of pain or humane intervention. This paper highlights the benefits, caveats, and potential applications of grimace scales.

## 1. Introduction

The 3Rs, Replacement, Reduction and Refinement, is a fundamental framework used internationally to ensure the ethical and justified use of animals in research [1]. The implementation of refinements is required to alleviate and minimise the pain and suffering of animals used in research. Public acceptability of animal use in research is contingent on satisfying the ethical and legal obligations to provide appropriate pain relief along with humane endpoints for potentially painful procedures. To fulfil this obligation, staff, researchers, veterinarians, and technicians must rapidly, accurately, efficiently and consistently identify and assess signs of pain in their target species, and act accordingly. The ability to identify and assess pain and suffering is paramount to animal welfare in research to prevent undue suffering and any possible consequent negative impact on research outcomes.

Identification of pain may be based on several indicators such as physiological, behavioural, or physical ones. Each of these has been used to develop different pain scoring systems, with potential benefits and limitations in identifying and assessing pain. Grimace scores are a promising adjunctive behavioural technique in some mammalian species to facilitate in identifying and assessing pain in research animals. The use of this method can be beneficial to animal welfare and research outcomes by identifying animals that may require alleviation of pain or humane intervention. A discussion of the benefits of grimace scales, including their potential applications, is included in this paper.

### 1.1. Eligibility Criteria

The inclusion criteria were: publication in English; assessments of pain in research animals and livestock; mammalian grimaces scales in animals; studies key to the development of grimace scales; studies that used facial units as an indicator of pain assessments in animals.

### 1.2. Search Strategy

The search strategy aimed to only find articles published in English or translated into English. There was no restriction on the date of publication. Articles were searched for between June–July 2020. Keywords used to search all databases and references sources were animal grimace score, animal grimace scale, animal pain assessment, animal pain indicators, animal pain face, animal pain scales and the NC3Rs website. All papers were retrieved and downloaded into Endnote with X8.0.1 with any duplicates removed. What is pain and why does it matter?

The International Association for the Study of Pain defines pain as: ‘An unpleasant sensory and emotional experience associated with, or resembling that associated with, actual or potential tissue damage’ [2]. Pain can be further categorised as acute, visceral or chronic. Acute pain serves an evolutionary and adaptive function to signal and avoid potential or actual damage to tissues. This type of pain may result from an injury or surgical wound [3,4,5]. Visceral pain is due to the activation of stretch or pressure receptors in visceral organs. Unalleviated or poorly treated acute pain can progress to chronic pain. The latter is the result of neuroplastic changes occurring within the nervous system, rendering the body more sensitive to pain and can even create sensations of pain without any external stimuli [3,4,5,6,7]. It is important to be able to manage and assess all types of pain in research mammals and avoid the inadvertent development of chronic pain. While the types of pain may manifest differently, research staff must be able to assess and alleviate pain to maintain optimal animal wellbeing (mental and physical).

There are moral, legal and ethical obligations that require those working with animals to manage pain [5,7,8,9,10,11,12,13]. The recognition, assessment and treatment of pain is an essential aspect in public support and acceptability in the use of animals for research [7,12,13]. Using the precautionary principle, animal ethics committees and research staff must acknowledge the potential for pain [5,10,12,14]. They must also consider the experimental and animal welfare consequences of pain and take steps to ensure pain is adequately managed during procedures [3,4,5,15]. Regulatory frameworks often apply, as a precaution, the anthropomorphic principle, by which any procedure causing, or expected to cause, pain in humans, may produce pain in animals. The European Union, United Kingdom and Australian regulations operate on this principle and require the alleviation of pain for research animals. Exceptions to the alleviation of pain may be granted for studies that measure pain and/or distress. However, even in these exceptional cases, there is always a maximum threshold level of pain before intervention is required [5,10,12,14,15,16,17]. Although management of pain and associated humane interventions will vary due to the nature of the experimental outcomes, researchers are required to intervene with predetermined criteria to alleviate pain or, if necessary, humanely euthanise animals. For practical purposes, animal users can only fulfil this obligation for humane intervention if they are able to identify pain rapidly, consistently and accurately in their target species.

Unalleviated pain results in alterations in animal behaviour, physiology, and physical states [18,19,20]. These changes can be identified in various ways through behavioural observation, biochemistry, haematology, endocrinology and physical alterations in locomotion or posture [4,21]. In addition to the suffering of the animal, these changes may impact experimental outcomes and become a confounder by increasing experimental variability and producing negative affective states. Conversely, positive emotional states of animals are linked to less experimental variability and more robust experimental results [22,23]. The full spectrum of potential confounders to unmitigated pain is not entirely understood; however, the literature supports that there are experimental and animal welfare benefits in identifying and subsequently treating pain and alleviating negative effects on animals used for research [3,17,19,21,22,23,24].

### 1.3. Pain Faces

Humans are known to display a series of facial expressions linked to the experience of pain [25,26]. Those so-called ‘pain faces’ are used in human medicine to detect, as well as assess, pain in non-verbal humans (i.e., infants) [7,25,26]. These pain faces can be used to develop grimace scales and capitalise on the human propensity to focus on the facial area [27,28]. The conservation of these pain faces is also present in many non-human mammalian species [26,29,30,31,32,33,34] and are a naturally useful method in the identification and assessment of pain. However, as with any technique, Grimace scales have benefits and limitations. These are important to acknowledge and take into considerations before their use.

### 1.4. Pain Assessment Requirements

There are a series of important considerations when determining if a method is an appropriate test in identifying and/or assessing pain. The testing method must reliably produce the same result independent of the observer and the number of times an animal is observed. These are, respectively, known as intraobserver and interobserver agreement. It should also be consistent between testing timepoints and observers [7,35,36]. An ideal method should be easy to train and not require specialist knowledge or equipment [7,33,36,37,38].

A suitable test must demonstrate validity by accurately determining or reflecting the presence or absence of pain [7,35,36,39]. To determine the validity of a pain assessment technique, we should test the animals before the painful stimulus, after the introduction of the painful stimulus and once pain relief has been provided. The test should demonstrate an absence of pain before the painful stimulus, an increase in pain at the introduction of the painful stimulus, and a subsequent reduction in observed pain on the delivery of an appropriate analgesic [7,36]. Ideally, the test should be able to demonstrate a dose–responsive curve to pain based on the administration of appropriate analgesia [40,41,42,43,44].

The specificity and sensitivity of a test are also crucial to ensure animals are correctly identified when pain or welfare concerns arise. If the specificity is too low, there is a risk of pain being incorrectly identified, potentially leading to unnecessary interventions such as pain relief or humane euthanasia [3,7,15,36]. Alternatively, if the sensitivity is too low, experimental animals may reach their threshold for intervention while being inaccurately identified as not painful, therefore remaining in pain possibly even beyond their humane endpoint. An appropriate method would demonstrate both high sensitivity and specificity to ensure correct assessment and correct management of arising pain or welfare issues [3,7,15,36].

Cage or pen-side pain identification techniques should rely on spontaneous rather than retrospective indicators of pain. It ensures humane intervention can be applied promptly with animals not left in distress for any extended length of time [9,45,46]. The assessment of pain should preferably be a non-invasive method, to avoid the risk of eliciting a pseudoanalgesic stress response, inhibiting the ability of the observer to detect pain accurately [47,48,49,50]. Techniques such as assessing the quality of nest-building in mice [3,15,18,51,52,53] or degree of burrowing in rodents are non-invasive, observatory, proxy measures to wellbeing and potentially pain [3,15,18,54,55].

### 1.5. Confounders to Pain Identification

Some caveats must be maintained when selecting a pain assessment technique. Many pain assessment indicators may be ambiguous. The choice of a pain identification tool or methodology must be specific to the species and validated for the procedures or experimental work being performed [56,57,58,59,60]. It well accepted that not all animals demonstrate the same signs of pain, even for a similar nociceptive stimulus [36,61]. Many research animals are prey animals and as such, are prone to hide signs of pain or demonstrate a freeze response, rendering pain assessments challenging [21,46,50,62,63,64,65,66]. The types of procedures or experiment performed should not obscure the ability of the technique to detect pain [39,56,57,58,67]. Pain identification should be consistent across the species regardless of sex, strain or breed; however, differences in pain thresholds between sexes or strains may exist [67,68]. Additionally, some natural behaviours (i.e., flehmen response, aggression) [3,4,32,69,70,71,72] or physiological indicators (i.e., cortisol, heart rate) [3,4,5,7,15,17,21,66] may be equivocal and require differentiation. Whenever possible, the choice of technique should accurately identify an animal in pain, independent of the procedure or behaviour performed, species, affective or physiological state, sex, strain or breed.

### 1.6. Non-Grimace Scale Pain Assessment

The individual expression, magnitude and experience of pain can vary between animals [67,68]. There are known difficulties in measuring the magnitude of a particular animal’s pain or distress which can make the absolute measurement or degree of pain challenging to assess [4,21,62,68,73,74,75].

Before the development and use of grimace scales, a variety of indicators have been used in an attempt to identify and assess pain. These can be grouped into behavioural, physiological and physical indicators (Table 1). Typically, behavioural indicators have the benefit of being non-invasive, observational, requiring limited equipment and offering an opportunity to capture signs of pain in species or individuals that may hide signs of pain (i.e., prey) [3,21,50,51,52,53,54,55,66]. Many behavioural assessment techniques take time (>5 min), require extensive training, are more retrospective than spontaneous, and may be non-specific proxy indicators to pain [4,18,76,77,78]. Physiological indicators (neuroendocrine or sympathetic nervous system) are often non-specific markers related to stress or distress [3,5,21,79,80]. They do have the benefit of being relatively quantifiable but often require specialised equipment, are retrospective, usually require animal handling or restraint with the potential for a confounding pseudoanalgesic effect [47,48,49] or are a non-specific stress response [3,4,8,81]. Physical indicators such as changes in posture, locomotion and production yields, have been correlated with the presence of pain in animals [3,8,15,17,21,64,66,82,83,84,85]. However, physical indicators are just as often non-specific indicators of non-painful animal wellbeing or environmental factors [3,5,8,15,17,21,64,66,82,84].

Ideally, a pain assessment technique should ensure accurate pain identification and minimal opportunity for the confounding of experimental outcomes due to experimental procedures, sex, breed/strain, or species. At present, there is not a single non-invasive, low-cost behavioural, physical or physiological pain assessment technique that is spontaneous, pain-specific, easy to train and quick to use (Table 1) [5,15,16,21,38,46,53,78,86,87,88,89]. With the exception of some behavioural ethograms [63,78,90,91], other methods are unable to give a reliable dose-dependent response to pain. While many pain identification methods have their use and benefits, their use in the cage or pen-side management of animal pain and/or in the timely and appropriate application of humane intervention is limited.

Thus, a myriad of techniques has been developed in an attempt to assess and capture the various expressions of pain in animals. These tools usually revolve around three dimensions: behavioural, physiological, and physical [5,73]. Table 1 categorises and reviews some commonly used assessments [3,4,7,8,15,21,80] in terms of their dimension, ability to be timely (spontaneous), non-invasiveness, spontaneous, easiness to train, and low-cost with minimal or no equipment requirements.

**Table 1 animals-10-01726-t001:** Behavioural, physiological, and physical pain assessment techniques.

Category	Assessment	Non-Invasive	Easy Training	High Cost	Special Equipment	Time Required >5 min	Spontaneous	Publications
Behaviour	Ethogram	Y	N	N	N	Y	Y	[4,5,7,15,17,18,21,52,63,66,78,86,87,91,92,93]
	Nesting	Y	Y	N	N	Y	N	[3,15,18,51,52]
	Burrowing	Y	N	N	N	Y	N	[3,15,18,54,55]
	Vocalisation	Y	Y	Y	Y/N *	Y	Y	[4,5,8,17,21,45,66,68,81,88,93,94]
	Grooming	Y	Y	N	Y	Y	N	[3,4,5,8,15,17,18,21,52,66]
	Real-time Grimace Score	Y	Y	N	N	N	Y	[37,54,55,70,85,92,95,96]
Physiological	Heart Rate or Respiratory Rate	N	Y	N	Y	N	Y	[3,4,5,15,17,21,66,93]
	Biochemical marker	N	N	Y	Y	Y	N	[3,4,15,17,21,93]
Physical	Weight loss or failure to gain weight	Y	N	N	Y	N	N	[3,5,8,15,17,21,64,66]
	Reduction in production **	Y	N	N	Y/N ***	Y	N	[21,66,84]
	Lameness	Y	Y/N ****	Y	N	N	Y	[3,5,15,17,21,37,66,82,85,93]
	Postural change	Y	N	Y	N	N	Y	[3,4,5,9,15,17,58,83,93]

* Ultrasonic vocalization monitoring requires special equipment; ** e.g., Egg or Milk; *** Eggs can easily be counted; **** Obvious signs of lameness are easy to train but more subtle lameness may be more difficult.

### 1.7. Grimace Scales in Animals

Grimace scales are proving to be a useful methodology for the identification of pain in research that meets most of the prerequisites for identifying and assessing pain in research animals. A range of research species-specific grimace scores has been developed (Table 2) and used in a wide range of experimental studies and research settings (Table 3). The initial methodology in the mapping of pain and the development of a facial action coding system (FACS) was developed in humans [97,98]. A FACS is an anatomical classification system used to map facial movements and facial muscles areas involved in facial contraction and relaxation. Photographs and videos scored by blinded observers serve as the base of facial mapping for FACS. FACSs offers the ability to code and identify expressions of pain via the individual components of facial expressions known as facial action units (FAUs) [99]. FAUs consistent with the expression of pain can then be used to develop a pain face or ‘grimace’ [99]. Regions of the face that have been found to change during the expression of pain include the eye, nose, cheek, mouth, ear and whiskers [8,81,100]. The position or carriage of the head is also found to change in some species as well [33,64,68,85,101,102]. The FAUs related to the expression of a grimace face in mammalian animals used in research are included in Table 4. From this known ‘grimace face’, the severity of the pain experienced can be objectively scored from images and/or film of animals in a known naturally (i.e., lameness, mastitis) [37,82,85] or experimentally induced (i.e., plantar incision [41,44,103] state of pain.

Table 2 summarises many of the available studies that demonstrate a successful use of grimace scales in research animals. The table outlines which species-specific grimace scales have been validated, shown to be pain-specific, demonstrated a dose-dependent relationship, used in real-time and were easy to use. The different pain states to which they are applicable is also listed. In all but one species (guinea pigs) [63,78,90], observers were found to correctly, reliably and objectively identify pain in animals when using facial expressions or facial action units.

Control animals (negative or positive) were also included throughout this process and a simple species-specific grimace scale was developed [25,33,41,107]. The scoring system most commonly used in grimace scales is a three-point scale to determine if a specific FAU is not (score = 0), moderately (score = 1), or obviously present (score = 2) [41,45,100]. The scale must then demonstrate a dose-dependent change in pain scores on the delivery of analgesia [7]. Further research is typically performed to ensure the applicability of the grimace scale across multiple pain scenarios or environments, sex, strain/breed, age, as well as type and length of painful stimuli [7]. The scoring system can be used three ways. Firstly, it can determine either the absence or presence of pain. Secondly, it can offer some distinction between the intensity of pain via the summation of total scores. A change by two or more points is considered to be a legitimate alteration in pain intensity [133]. Thirdly, a threshold score can be set to offer guidance to research staff as to when to intervene to provide pain alleviation or humane euthanasia of research animals. The process of developing a grimace scale is time intensive but once developed and validated is relatively easy to train research staff to use [7,38,70,85,101].

## 2. Advantages and Uses

Grimace scales have been applied across numerous research models, species and environmental contexts [41,128] (Table 4). They are a technique that can also be used to detect pain in existing pain research models as well as analgesic drug studies [40,41,42,45,60,77,109,110,128,129,130]. Grimace scales offer the ability to detect and assess the severity of pain, determine the potential benefit of any analgesic intervention and assist in identifying humane interventions. The technique is of practical value as it can be used at the cage or pen-side level as a spontaneous indicator of pain [39,41,55,75,92]. As a methodology, it has the added benefit of being easy to teach to a range of observers including research staff, clinical veterinarians, animal scientists and undergraduate and graduate students [38,41,55,75,129]. Overall, the grimace scale methodology appears to be acceptably conserved and validated across a number of mammalian species and range of experiments. It is likely this technique has the capacity to be applied across an even greater range of mammalian species and experimental settings (Table 3 and Table 4). However, a careful systematic assessment will always be required to ensure applicability, accuracy and validity.

Grimace scale facial expressions are proving to be a useful [81] complement to existing tools in the assessment of animal wellbeing. The scores generated from the grimace scale should be used in conjunction with the context in which the animal is scored, its history, the procedure performed and the general parameters for wellbeing and signalment (sex, strain, species). When used appropriately, it is an excellent method to identify pain and as an adjunct to maintaining animal wellbeing in research studies [3,64,70,82,85,87]. Using this technique has the potential to improve pain detection in research animals and enable observers (i.e., research staff) a better opportunity to provide analgesia, humane euthanasia or identify animals requiring reassessment. The use of these grimace scales can be a vital tool to enable mitigation of the experience of pain in animals and refine animal welfare outcomes [41,60,66,75,76,82,100,114,128]. Unlike other types of pain assessment, grimace scales are spontaneous and usable in real-time [7,45,55,76,87,91,92,101]. They can also be matched and corroborated against other known indicators of pain or painful diseases including, but not limited to, lameness [37,64,82,85], cortisol [70], behavioural ethograms [81,85,91,92], acute laminitis [37], mastitis and foot rot [82]. A future area for development and benefit is the use of software automation in the development and scoring of facial expressions. The use of scoring software along with the installation of video cameras into enclosures may be able to enhance and hasten the development of grimaces, offer highly accurate grimace scores for animals in pain but also allow the remote monitoring and scoring of affected animals [41,59,134,135].

Another benefit of the system is simplicity, as it enables staff to distinguish a painful face from a non-painful one. Using a three-point scale is thought to be very useful in the reduction in subjectivity and offers observers greater clarity, confidence and support as to when to administer pain relief or humane intervention [7,75,81]. Reduction in grimace scores has been shown to occur on the application of pain relief [33,35,41,45,82,85,100,102] in a dose-dependent manner [40,41,128]. Therefore, grimace scales have the potential to assess both the presence and severity of pain. The use of grimace scales can alert research staff to animal discomfort, which may require additional monitoring, assessment or analgesia.

Grimace scales are a non-invasive method in the detection of pain [7,81,100]. Many of the animals utilised in research are known ‘prey’ species with a high degree of stoicism and evolutionary adaptation to minimise expressions of pain or poor welfare states [50,63,64,65,74]. Consequently, an ideal pain identification and assessment should be non-invasive and should reduce the possibility for these prey animals to minimise their expression of pain or for the potential of stress-induced analgesia [47,48,49,50].

Both experienced and inexperienced observers can identify pain with a significant intraobserver and interobserver agreement [41,57,60,66,76,82,100,114]. A potential benefit of using grimace scales to identify and assess pain in animals is that extensive animal experience is not required. Observers varied in their background experience to research and animal work and their training in pain assessment techniques. The observers ranged from students (undergraduate and postgraduate), veterinarians, animal care professionals, and early to late-career researchers [33,38,70,75,101,106,114,116]. Another favourable outcome when using grimace scales is that a natural empathy or innate understanding of animal behaviour is not necessary nor is a belief in the ability of an animal to experience pain. Through the use of a grimace scale, pain identification and assessment can be more objective (for or against the presence of pain). It also requires research staff to formally record a score and monitor animals for signs of pain, which can offer a more precise framework to determine when humane intervention or pain relief is needed [36].

The apparent usefulness of grimace scales could be related to several factors. One of which is that it capitalises on the innate human tendency to focus on the facial area when observing an animal [28]. Interestingly, many FAUs (orbital tightening, ear position and cheek area) appear to be conserved across mammalian species [33,41,45,82,100] (Table 3) and may be tapping into an evolutionary conservation repertoire of known FAUs. It may help explain how even the single identification of a few potentially evolutionarily conserved FAUs can still be useful in detecting pain [35]. It is supported via statistical modelling which has identified the FAUs most strongly correlated with a pain face, thereby offering the potential to isolate which FAUs are critical for use in a grimace scale (i.e., statistically significant) and which ones may detract from the scale (i.e., equines have four and mice have two critical FAUs) [35]. It may explain why grimace scales are one of the few techniques proven to be robust across several different mammalian species when compared to other pain assessment techniques [34]. However, by using only the minimum number required of FAUs to score pain, the ability to determine appropriate intervention thresholds and assess pain intensity may be reduced.

The use of FACS and subsequent combinations of FAUs appears to be an excellent method to identify changes in facial features, which are consistent with the experience of pain in animals [33,40,41,76,82,85,92,101,104,106,114,130]. The grimace scale method seems to meet many of the requirements for an ideal pain identification technique. It is known to be a reliable and validated method of assessing pain in many of the commonly used research animals [33,35,40,41,45,70,82,85,100,106,114].

## 3. Limitations

Similar to any tool, grimace scales have their caveats and limitations. The creation of pain and grimace scales takes considerable time to develop [3,41,61,62,99]. FAUs can be species-specific with each FAU requiring validation and ideally, statistical modelling and weighting, to determine its significance in the system [8]. This means the number of FAUs can vary amongst species with mice having five FAUs [100], rats four [41], sheep three to five [70,82], lambs five [107], equines six [37], ferrets three [106], cattle three to four [68,85], rabbits five [45] and pigs and piglets three [91,114]. While some FAUs movements appear to be tightly conserved (i.e., orbital tightening), others vary amongst species. These variations can be contradictory between species and may be due to age of the animal [107,115] and/or musculature of the face. The nose and philtrum areas tend to be areas with greater variation amongst mammals [37,82]. For example, rats and rabbits [41,45] will flatten their nose when in pain while mice and ferrets bulge their noses [100,106]. Therefore, each species requires the development of its own precise facial or grimace scoring system. Currently there are several commonly used mammalian research species that either do not have developed grimace scales or are yet to be fully developed. These include hamsters, dogs, guinea pigs, and non-human primates. Further work is needed to develop and determine the validity of grimace scales in these species.

Pain expression and threshold levels can also vary slightly amongst breed or strain [67,68]. Baseline grimace scores need to be taken for every cohort daily for approximately three days before the initiation of an experiment or potentially painful stimuli to minimise these variations [75]. False positives are known to occur in a small range of scenarios such as sedation/anaesthesia, sleeping status [41,56,75,100], or during bouts of aggression [32]. Therefore, grimace scales should not be used during those times. Additionally, it is important to note that facial variations may occur between individuals. As a result, absolute scores may be less important than a change in the score by two points or more (i.e., trends) [133], and a more ‘trends-based’ approach could be more useful. There are also times when the grimace scales can result in a false positive with animals not demonstrating a pain face during a known painful procedure. For example, ear clipping in mice did not demonstrate any changes in grimace scores [57] and neither did experimentally induced gastrointestinal mucositis in rats [58]. There is discussion around the differences found in the length of time of post-painful stimuli that an animal may display a pain face and hence a grimace score. Early peer-reviewed publications questioned the ability of grimace scales to be useful for more than 24 h after a painful event [59,100]. More current studies have demonstrated that pain can be identified in animals via grimace scales for more than 24 h and more than 14 days after a painful stimulus [55,82,108,120,128,129]. From the recent literature and available publications, it is clear this technique has applications beyond its initial use.

The history of the animal, the species, breed/strain, environmental context, procedures performed, and general parameters of wellbeing must be considered when using grimace scores [7,64,75,81]. There is still research to be conducted to explore the use of grimace scales. Currently, not every grimace scale has been fully validated (ferret, piglet, lamb) and additional species may yet benefit from their development (goats or other small mammals). Preliminary work does suggest that guinea pigs do not appear to be good candidates for facial pain scales. These studies used behaviourial ethograms which included elements of commonly and strongly conserved facial expressions (i.e., orbital tightening) and did not find any significant correlation of these expressions as indicators of pain [63,78,90]. It may be that grimace scales are not appropriate for these species or the FAUs associated with pain are different to other mammalian species. Many scales have only been used in specific settings or studies and need further work to determine if they are affected by common agricultural or animal procedures such as restraint in lambs [107] or piglets [115]. There is still variability in the available literature as to the length of time a grimace score can be detected in some species and studies [58,128] as well as its applicability of use [56] which should be further explored. While grimace scales have been developed and validated for several mammalian species, it is known there are species-specific variations in the expression of pain faces (guinea pigs) which may determine the development of a grimace to be unsuitable or require a different approach.

## 4. Application and Summary

In an ideal situation, a single pain identification technique would be sufficient across all species and scenarios; however, this currently does not exist and may never exist, given pain is an individual multifactorial experience [61]. Nonetheless, the growing body of literature is demonstrating that, overall, pain faces in mammalian species are often expressed and can be identified during most procedures, pain types, and contexts. Most of the variation found when using grimace scores to identify and assess pain is in the strength of association, the magnitude of certainty and the consistency of grimace score expression. Even with these variations, the use of grimace scales appears to be good at detecting pain in mammals [41,45,64,70,85,87,91,100,101,102,106,114]. However, if studies could be more standardised in their approach and the use of grimace scales, this may be beneficial in reducing minor confounding elements (i.e., handling, conspecifics) or in identifying areas of improvement. Future studies and the day to day practical and experimental application of this technique would benefit from having a formally validated and consistent training program, complete with video and photographic materials. A standard training program would be useful for grimace score users and has been useful for other pain scoring systems [38,46,86]. Part of the development, training and implementation of grimaces could be enhanced by the use of various technologies such as automated or semiautomated software for scale development and scoring via video surveillance [41,59,134,135]. These nascent technologies are often unfeasible due to cost, infrastructure constraints and a lack of development but in the future their use may play a greater role in grimace scoring systems.

The identification and mitigation of pain fulfil an essential and required aspect of refinement when working with animals in research. As of yet, no single indicator or technique is considered sufficient in the identification and assessment of pain. Several peer-reviewed publications have advocated multiple measures of animal welfare, and pain should be employed to mitigate the potential negative effects of pain on animal welfare and research outcomes [3,4,9,15,21,61,64]. Using a combination of relevant retrospective and spontaneous techniques applied on a case by case basis can maximise the opportunity to detect and assess pain in research animals. It minimises the chance for pain to be undetected and maximises the opportunity to preserve animal welfare and research outcomes. While there are known limitations, grimace scales to at least identify potential indicators of pain are useful tools [60]. The use of grimace scales with other parameters of pain and/or animals wellbeing is likely to increase the ability of research staff to identify and assess pain in mammals and offer appropriate humane interventions. At this time grimace scales are a potentially promising and important pain identification tool; however, further work should be performed in a consistent manner to validate existing work as well as explore new applications to other species, conditions and experimental studies.

To achieve good animal welfare and research outcomes and meet legal and ethical obligations, it is paramount to utilise a consistent and accurate pain identification method. The use of a grimace score can assist in fulfilling these obligations by identifying pain and allowing a timely intervention via analgesia or humane endpoints. Grimace scales are thus proving to be a valuable tool with a myriad of applications. Their use can offer improvements in animal welfare and more robust animal research outcomes [9,64]. While grimace scales are not without limitations, there is a growing body of literature and evidence to suggest they can be a significantly useful adjuncts in the detection and assessment of pain in a variety of species and research studies [7,35,41,66,76,100]. When used correctly by trained individuals along with an animal’s history and basic wellbeing criteria, grimace scales can be a practical, accurate and easy method to identify pain in research animals to provide refinements in experimental animal welfare and outcomes [38,61,75]. Future applications of their use could focus on different types of experimental studies, new species, neonates, standardisation in training protocols, and correlation of multiple observations over time.

## 5. Conclusions

While there are some identified limitations, grimace scales appear to be a valid tool for pain assessment in many mammalian animals, and have many benefits compared to non-grimace pain assessment techniques. Due to the simplicity of spontaneous use, non-invasive application, repeatability of results, interobserver and intraobserver reliability and ease of training, the use of grimace scales should be more frequently considered as an important parameter of interest in research and animal wellbeing. In addition, this technique has the capacity to satisfy the requirement for refinement in accordance with the 3Rs. Additional research into the use of grimace scales is required for other species, pain-related or other specific studies, and further validation.

## Figures and Tables

**Table 2 animals-10-01726-t002:** Summary of species-specific available grimace scores.

Species	Validated *	Specific To Pain	Dose-Dependent Relationship	Real-Time	Easy To Train	Acute Pain	Chronic Pain	Visceral Pain	Publications
Cattle	Y	Y	Y	Y	Y	Y	Y	N/R	[7,8,64,68,85]
Equine	Y	Y	Y	Y	Y	Y	N/R	Y	[33,35,37,39,87,92,102,104,105]
Feline	Y	Y	N/R	Y	Y	Y	N/R	N/R	[101]
Ferret	N/R	Y	N/R	Y	Y	Y	N/R	N/R	[106]
Guinea Pig	N/R	N/R	N/R	N/R	N/R	N/R	N/R	N/R	[63,78,90]
Lamb	Y	Y	N/R	Y	Y	Y	N/R	N/R	[107]
Mouse	Y	Y	Y	Y	Y	Y	Y	Y	[3,35,40,57,67,76,77,83,100,108,109,110,111,112,113]
Pig	Y	Y	N/R	Y	Y	Y	N/R	N/R	[114]
Piglet	Y	Y	Y	Y	Y	Y	N/R	N/R	[91,115,116,117]
Rabbit	Y	Y	Y	Y	Y	Y	N/R	N/R	[24,28,45,96]
Rat	Y	Y	Y	Y	Y	Y	Y	Y	[41,42,43,44,55,60,94,95,103,113,118,119,120,121,122,123,124,125,126,127,128,129,130]
Sheep	Y	Y	Y	Y	Y	Y	Y	N/R	[70,82]

* Validated by reduction in grimace scores on receipt of pain relief and/or corroborated with other pain behaviours or testing; N/R denotes lack of publications available.

**Table 3 animals-10-01726-t003:** Grimaces Scales Facial Action Units by Species.

Facial Action Unit or Indicator	Species
**Orbital Tightening and/or Change in Orbital Area**	Cattle [68,85] Equine [33,102] Feline [101] Ferret [106] Lamb [107] Pig [114] Piglet [116] Mouse [100] Rabbit [45] Rat [41] Sheep [70,82]
**Cheek Tightening or Flattening**	Cattle [68,85] Equine [33] Lamb [107] Pig [114] Piglet [116] Sheep [70,82] Rabbit [45] Rat [41]
**Cheek Bulge**	Ferret [106] Mouse [100]
**Nose Bulge**	Ferret [106] Mouse [100] Pig [114] Piglet [116] Rabbit [45]
**Nose Flattening**	Equine [33] Lamb [107] Rat [41]
**Lowered Head Carriage**	Equine [33,102] Cattle [68,85] Feline [101] Sheep [70]
**Lip curling**	Equine [92] Sheep [70]
**Abnormal Nostril or Philtrum shape**	Cattle [68,85] Equine [33,102] Lamb [107] Rabbit [45] Sheep [70]
**Eye Rolling**	Cattle [68]
**Ear Position**	Cattle [68] Equine [33,102] Feline [101] Ferret [106] Lamb [107] Mouse [100] Pig [114] Piglet [116] Rabbits [45] Sheep [70,82]
**Whisker Position**	Feline [101] Ferret [106] Mouse [100] Rabbit [45] Rat [41]
**Abnormal Lip or mouth shape**	Equine [33,102] Feline [101] Lamb [107] Sheep [70]
**Open Mouth +/− Tongue Extruded**	Cattle [68]

**Table 4 animals-10-01726-t004:** Grimace Scales by Experimental Study or Pain Type.

Pain or Study Type	Species
**Visceral**	Cattle [85] Equine [92] Mouse [100] Rat [44,55,121,123]
**Chronic**	Mouse [108] Rat [55,120,128,129]
**Acute**	Equine [33,35,37,102,105] Cattle Gleerup [68,85] Ferret [106] Lamb [107] Mouse [35,40,76,77,100,109,110,111] Pig [114] Piglet [91,115,116,117] Rabbit [28,45] Rat [41,43,44,55,60,95,103,118,120,122,123,124,125,126,127,128,129,130,131,132] Sheep [70,82]
**Neuropathic**	Mouse [108] Rats [95,126]
**Soft Tissues Surgery**	Equine [33,35] Ferret [106] Lamb [107] Mouse [40,76,77,111] Pig [114] Piglet [115,116,117] Rabbit [28] Rat [41,44,60,125,127,130,132]
**Orthopaedic Surgery**	Mouse [109,110] Sheep [70]
**Surgical, Mechanical, Branding,** **or Hypersensitivity Injury**	Equine [102] Cattle [68] Mouse [100,112] Rabbit [45] Rat [41,103]
**Dental**	Equine [104,105] Mouse [112] Rat [43,124,128,129,131]
**Stifle injury**	Mouse [100] Rat [120]
**Intraplantar CFA**	Rat [41,103,118]
**Intracerebral Haemorrhage**	Rat [42]
**Head and Ocular Pain**	Mouse [109] Equine [105]
**Footrot**	Sheep [82]
**Mastitis**	Cattle [85] Sheep [82]
**Lameness**	Cattle [85] Sheep [44,82,118,130,132]
**Sickle Anaemia**	Mouse [83]
**Cold hypersensitivity**	Mouse [83]
**Myocardial Infarction**	Mouse [110]
**Laminitis**	Equine [37]
**Cystitis**	Mouse [100]
**Sepsis**	Rat [121]

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
