# Peer review of "Grimace Scores: Tools to Support the Identification of Pain in Mammals Used in Research"

_animals, 2020, doi:10.3390/ani10101726_

Round 1

Reviewer 1 Report

In their manuscript, Cohen and Beths review the extant literature studying pain assessment techniques in general, and grimace scales specifically, in animal research. They describe in detail the requirements for an appropriate method for pain assessment and summarize the advantages and limitations of Grimace Scales. They conclude that the Grimace Scale is a promising tool for pain assessment assuming it is seen in the right context.

The article is of great interest to the readers as it gives a very good overview of the existing grimace scales in different mammals used in research. On the other hand, it also points out a responsible use of this promising tool.

I just have a few comments:

  1. How did you select the literature on which the article is based? Did you use specific search terms and/or literature databases?
  2. A new approach is to automate the grimace scale. Is there a reason why you do not go into this aspect in your article?
  3. The last sentence of the abstract (page 1, line 37) as well as the last sentence of the introduction (page 2, line 59) is identical. Perhaps you can find a slightly different formulation.

Author Response

Dear Peer Reviewer, 

Thank you very much for your excellent and thoughtful questions and feedback. Please see the below answers and let us know if we are able to provide further information to assist in answering your queries.

1. How did you select the literature on which the article is based? Did you use specific search terms and/or literature databases?

Search terms used were: pain assessment, grimace scale, grimace score, pain face, pain scales and the NC3R's website which has a list of many of the publications related to pain scores in rodents and rabbits. 

2. A new approach is to automate the grimace scale. Is there a reason why you do not go into this aspect in your article?

We did find literature and information relating to the automation of the grimace scales. As this is not widely available or practical as a real-time, cage-side tool at this time we had not included this information. However we have now added this as potential future area for development and discussion point as per the feeback. 

3. The last sentence of the abstract (page 1, line 37) as well as the last sentence of the introduction (page 2, line 59) is identical. Perhaps you can find a slightly different formulation.

Thank you for pointing this out. We have updated and refined the sentence. 

Reviewer 2 Report

In the submitted paper, the authors present a review of the published and validated grimace scales (GS) for 10 mammalian research species and even consider the systems at different life stages (Lamb vs. sheep; Piglet vs. Pig). The paper is overall well written with clarity and logical progression of the discussion. The tables (1-4) offer a concise visual summary of the cited material for comparison. 

Major point:

This is a review, not an original article.

Minor points for consideration:

As the paper focuses on research animals, it would be prudent to include specific information about common species that do not currently have GS's - hamsters, nonhuman primates, and dogs. This could be included under limitations, but could otherwise be framed as future directions within the field.

Similarly, new software programs/artificial intelligence are being developed to remove human subjectivity from these assessments, some discussion for future direction should be included. 

The negative results for guinea pigs should be expanded in discussion. Did it result from poor study design that should be repeated, was it a species specific issue that will never work, etc? Any additional species which might have a similar result?

Line 413 - correct double citation

Author Response

Dear Reviewer,

Thank you for your comments and feedback. We have endeavoured to address your comments in this manuscript. 

1. As the paper focuses on research animals, it would be prudent to include specific information about common species that do not currently have GS's - hamsters, nonhuman primates, and dogs. This could be included under limitations, but could otherwise be framed as future directions within the field.

2. Similarly, new software programs/artificial intelligence are being developed to remove human subjectivity from these assessments, some discussion for future direction should be included. 

3. The negative results for guinea pigs should be expanded in discussion. Did it result from poor study design that should be repeated, was it a species specific issue that will never work, etc? Any additional species which might have a similar result?

4. Line 413 - correct double citation